# Superfluid response of an atomically thin gate-tuned van der Waals superconductor

Alexander Jarjour [1], G. M. Ferguson[1], Brian T. Schaefer [1], Menyoung Lee[2,3], Yen Lee Loh[4], Nandini Trivedi [5] & Katja C. Nowack [1,2] ✉

A growing number of two-dimensional superconductors are being discovered in the family of exfoliated van der Waals materials. Due to small sample volume, the superfluid response of these materials has not been characterized. Here, we use a local magnetic probe to directly measure this key property of the tunable, gate-induced superconducting state in $MoS_2$. We find that the backgate changes the transition temperature non-monotonically whereas the superfluid stiffness at low temperature and the normal state conductivity monotonically increase. In some devices, we find direct signatures in agreement with a Berezinskii-Kosterlitz-Thouless transition, whereas in others we find a broadened onset of the superfluid response. We show that the observed behavior is consistent with disorder playing an important role in determining the properties of superconducting $MoS_2$. Our work demonstrates that magnetic property measurements are within reach for superconducting devices based on exfoliated sheets and reveals that the superfluid response significantly deviates from simple BCS-like behavior.

The two defining properties of a superconductor are a vanishing electrical resistance and the expulsion of magnetic fields below a characteristic critical temperature, $T_c$. Typically, superconductivity is first identified in a material by observing a sharp drop in the resistance at $T_c$. However, resistance measurements only give limited information about the superconducting state forming below $T_c$, and other experimental probes are needed to reveal its nature. Measurements of the strength with which the superconductor screens a magnetic field directly probe the superfluid stiffness, $\rho_s$, and have provided insight into the nature of unconventional superconductors[1–4]. From $\rho_s$, the superfluid density can be extracted, which in a clean BCS superconductor at $T = 0$ is expected to be equal to the normal carrier density[5]. Comparing the superfluid density to the normal carrier density, pair breaking by impurity scattering and other mechanisms can be identified[6]. In two-dimensional superconductors, the onset of $\rho_s$ may show fingerprints of the Berezinskii-Kosterlitz-Thouless (BKT) transition[7,8].

A growing family of atomically thin superconductors is realized by mechanically exfoliated sheets of van der Waals (vdW) materials. These include two-dimensional (2D) superconductors based on bulk superconducting materials such as $NbSe_2$[9,10], $NbS_2$[11], and $TaS_2$[12], as well as 2D superconductors that are induced by electrostatic gating such as $MoS_2$[13], $WS_2$[14,15], $MoTe_2$[16], $WTe_2$[17,18], twisted bilayer graphene[19], and ABC stacked trilayer graphene[20]. A variety of superconducting phenomena have been observed in atomically thin vdW superconductors, such as robustness against large in-plane magnetic fields[21–23], superconductivity in the vicinity of correlated electronic states[19,20], a dramatically enhanced $T_c$ in the monolayer limit[12,16], and unusual symmetry breaking in the superconducting state[24]. A detailed study of the transport properties of $NbSe_2$ with varying thickness has shown that dissipationless transport is highly fragile to temperature, applied magnetic field and the employed bias current[25] further highlighting the need for directly probing the phase coherence of the superconducting state in vdW materials. However, due to the typically small sample size, only a few measurements beyond electronic transport which directly

[1]Laboratory of Atomic and Solid State Physics, Cornell University, Ithaca, NY, USA. [2]Kavli Institute at Cornell for Nanoscale Science, Ithaca, NY, USA. [3]School of Electrical and Computer Engineering, Cornell University, Ithaca, NY, USA. [4]Department of Physics and Astrophysics, University of North Dakota, Grand Forks, ND, USA. [5]Department of Physics, The Ohio State University, Columbus, OH, USA. ✉e-mail: kcn34@cornell.edu

probe the superconducting state below $T_c$ are available[26,27], and no characterization of the magnetic response has been reported for any atomically thin vdW superconductor.

Here, we report direct measurements of the magnetic response of the gate-induced superconducting state in few-layer MoS$_2$. Although MoS$_2$ is a semiconductor when undoped, ionic liquid gating can induce an electron accumulation layer at the surface of a MoS$_2$ flake which exhibits superconductivity at carrier densities exceeding $0.5 \times 10^{14}$ cm$^{-2}$ [13]. $T_c$ changes non-monotonically with the carrier density with a maximum of approximately 10 K. Superconductivity is retained in the monolayer limit[28], but is always in the 2D limit regardless of the flake thickness because the accumulation layer is approximately confined to the topmost layer[21,29]. Spin-valley locking in the electronic bandstructure of MoS$_2$ gives rise to Ising protection of the superconducting state leading to an in-plane critical field dramatically exceeding the Pauli limit[21,29]. Recently, tunneling measurements have suggested that the order parameter is not fully gapped[26], a possible signature of an unconventional superconducting state.

## Results

### Local magnetic measurements using scanning SQUID

We fabricate our devices from exfoliated MoS$_2$ flakes with a thickness of 3–10 layers, and pattern them into disks with Ti/Au contacts to allow gating and electrical transport measurements (Fig. 1b, all devices shown in Supplementary Fig. 1). In previous work on superconducting MoS$_2$, droplets of ionic liquid were used to induce the required carrier densities[13]. To bring a local probe sufficiently close to our devices, we use a spin-coated, ~2-μm-thick ionic gel. The ionic liquid used in previous work makes up >90% of the gel, by mass. In addition, we apply a

backgate voltage, $V_{BG}$, across 300-nm thick SiO$_2$ to our devices (a diagram of this dual-gate setup is shown in Supplementary Fig. 1a). Figure 1a schematically shows how we measure the magnetic response of a device. A scanning superconducting quantum interference device (SQUID)[30,31] with a pickup loop and a concentric field coil is centered above the device. Here the pickup loop and field coil have an inner diameter of 1.5 μm and 8 μm, respectively. An AC current in the field coil produces a small magnetic field, and we measure the resulting flux in the pickup loop using a lock-in amplifier. Away from the sample, this signal corresponds to the mutual inductance between the pickup loop and field coil. When the pickup loop/field coil pair is brought close to the device, the magnetic response of the device appears as a change in the mutual inductance. By measuring this change, we directly probe the magnetic response $\chi$ of the device.

Specifically, a superconductor generates currents to screen the applied magnetic field. The strength of the screening currents can be related to the superfluid stiffness $\rho_s$, the Pearl length $\Lambda$, and the superfluid density $n_s$ which are connected through $\rho_s = \hbar^2/(2\mu_0 k_B e^2 \Lambda)$ and $n_s^{2D}/m^* = 4k_B \rho_s/\hbar^2$, where $\hbar$ is the reduced Planck's constant, $\mu_0$ is the permeability of free space, $k_B$ is the Boltzmann constant, $e$ is the elementary charge, and $m^*$ is the effective mass. Based on the signal magnitudes we observe, our devices are in the weak-screening limit, i.e., $\Lambda \gtrsim R$, where $R$ is the device size (see Supplementary Discussion 1). In this limit, the magnetic response $\chi$ is directly proportional to $\rho_s$: $\chi = M_{geo}\rho_s$. The proportionality factor $M_{geo}$ depends on the SQUID dimensions, its height above the sample, and the device dimensions. We can model $M_{geo}$ to extract absolute values of $\rho_s$ and $\Lambda$ from our measurements (as described in Supplementary Methods 2). The estimate of $M_{geo}$ has systematic uncertainties due to uncertainties in the SQUID height, the exact device dimensions, and other geometrical factors; however, relative changes of $\rho_s$ as a function of $V_{BG}$ and temperature are captured with high accuracy. Figure 1c shows an image of $\chi$ at a constant height. From this, we identify the center of the device where we position the SQUID. We then measure $\chi$ at a fixed height as a function of temperature and $V_{BG}$ and simultaneously record the sheet resistance $R_\square$. To ensure that our data reflects the linear response of our devices, we vary the current in the field coil and confirm that the signal changes linearly with the current (see Supplementary Discussion 1). We estimate the root-mean-square AC applied magnetic field strength at the center of Device A and B to be 13 and 19 μT, respectively, and estimate that the screening current densities that flow in response are significantly smaller than critical current densities reported for superconducting MoS$_2$[28,32] (See Supplementary Discussion 1). Finally, we note that the size and height of our probe are comparable to the device dimensions. Therefore, our measurements probe a significant volume fraction of the device.

### Characterization of the superfluid response and resistance

Figure 2 shows $R_\square$ and $\chi$ as a function of temperature and $V_{BG}$ for two devices, which we label A (20-μm diameter) and B (15-μm diameter). An additional device is shown in Supplementary Fig. 3. A weakly temperature-dependent magnetic response from the ionic liquid has been subtracted from $\chi$ (shown in Supplementary Fig. 2). For both devices, $R_\square$ drops sharply as we lower the temperature, and the superfluid response appears when the drop in resistance is completed. At $V_{BG} = 0$, device A has a lower $T_c$ of ~6 K, compared to ~9 K in device B. For both devices, the critical temperature changes with $V_{BG}$. In device A, the resistive transition remains ~1.5 K wide across the backgate range, whereas in device B the transition broadens with decreasing $V_{BG}$ and changes shape. Likewise, we observe that the shape of the superfluid response versus temperature for device A is qualitatively independent of $V_{BG}$, but does vary for device B. However, in both devices, the response measured at the lowest temperature increases monotonically with $V_{BG}$. Finally, the resistance below the transition is finite for both devices at low values of $V_{BG}$. The simultaneously observed superfluid

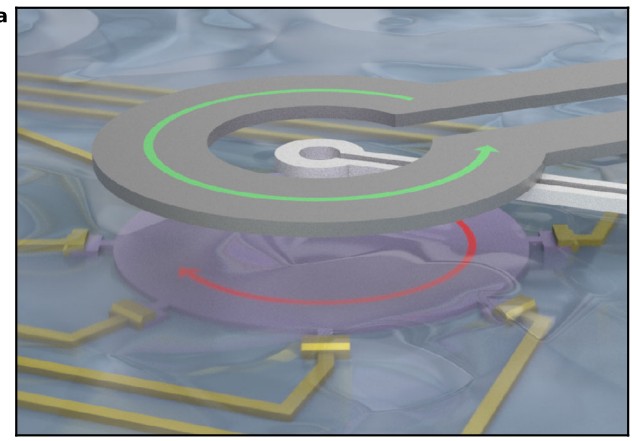

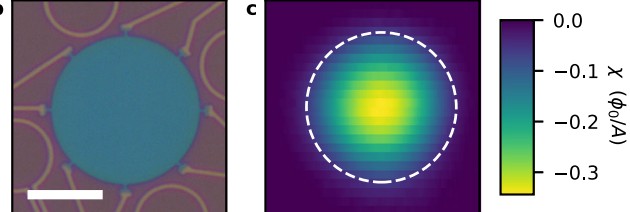

**Fig. 1 | Magnetic measurement of ionic gated MoS$_2$. a** A flake of MoS$_2$ (shown in purple) on a SiO$_2$/Si substrate is patterned into a disk and covered by a spin-coated ionic gel. The SQUID pickup loop (shown in silver), with concentric field coil (shown in dark gray) is approached to the sample. A current in the field coil produces a magnetic field, which results in an opposing screening current in the superconductor. The strength of the screening current is magnetically detected by the pickup loop. **b** Optical image of device A, a 20 μm diameter circular MoS$_2$ device (blue) with electrical contacts. The scale bar is 10 μm. **c** Image of the magnetic response of the device shown in **b** at 4 K. The white dashed circle has a diameter of 20 μm and indicates the device circumference.

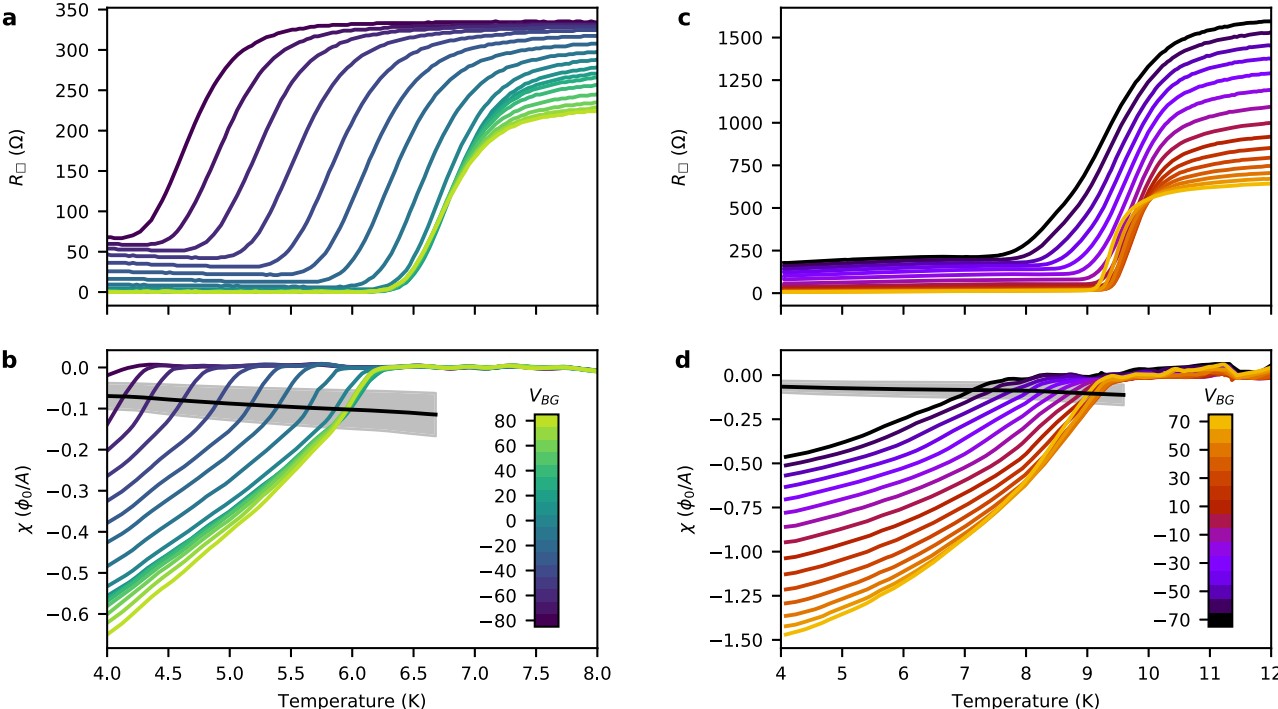

**Fig. 2 | Temperature dependence of the gate-tuned resistance and superfluid response. a** Sheet resistance, $R_\square$, and **b** superfluid response, $\chi$, of device A versus temperature. Different colors correspond to different backgate voltage $V_{BG}$ as indicated by the color bar in **b**. The black line corresponds to the universal BKT condition $\rho_s = 2T/\pi$. The gray shaded area indicates uncertainty in the universal condition arising from uncertainty in the SQUID height. **c, d** Same as in **a, b** but for device B.

response of the samples raises the question of whether this resistance is intrinsic to the superconducting state. While our ionic gel differs from the typically used ionic liquid, similar residual resistances have been observed previously in ionic-gated $MoS_2$[28]. A possible extrinsic explanation is a non-superconducting region along the periphery of the device, which would have a particularly pronounced effect in disk-shaped devices (see Supplementary Discussion 2). While determining the origin of the finite resistance is interesting, it is outside the scope of this work. In the following, we first focus on the superfluid response significantly below $T_c$, and then discuss the onset of diamagnetism close to the superconducting transition.

**Backgate dependence and the role of disorder**

In Fig. 3a, d, we summarize the backgate dependence of different temperature scales characterizing the superconducting transition. We plot $T_c^R$ at which $R_\square$ has decreased by 10% from the normal-state value and $T_c^\chi$ at which the superfluid response just rises above our noise floor. In Supplementary Fig. 9 we show these temperatures $T_c$ overlaid onto the data. To compare changes in the overall strength of the diamagnetic response versus $V_{BG}$, we plot values of $\chi$ at a fixed fraction of $T_c^\chi$ for both devices. Due to the comparably low values of $T_c^\chi$, we plot $\chi(.9\ T_c^\chi)$ for device A in Fig. 3b and omit values corresponding to the two lowest values of $V_{BG}$. In device B, $T_c^\chi$ is substantially higher across the backgate voltage range. Therefore, we plot $\chi(0.55\ T_c^\chi)$ in Fig. 3e. Lastly, we plot the normal-state $R_\square$ in Fig. 3c, f for devices A and B, respectively.

$T_c^R$ and $T_c^\chi$ both have a non-monotonic dependence on $V_{BG}$ for device A and B, which is also directly visible in the data in Fig. 2. In device B, the superconducting transition broadens as $V_{BG}$ decreases, which is reflected in the growing difference between the two temperatures. In contrast to the non-monotonic dependencies of $T_c^R$ and $T_c^\chi$, the superfluid response increases monotonically with increasing $V_{BG}$ doping, whereas the normal-state resistance decreases.

In the clean limit, all normal carriers are expected to condense into the superconducting state. Therefore, a monotonic increase in

the superfluid density and therefore $\chi$ can be expected as the normal carrier density, $n_n$, increases with $V_{BG}$. However, the observed change in $\chi$ over the full backgate range is larger than can be explained by a change of $n_n$ alone, especially in device B. We estimate the carrier density induced by the backgate as $7.0 \times 10^{10}\ cm^{-2}$ per volt given the 300 nm thickness of the $SiO_2$ with an approximate dielectric constant of 3.8. Our setup is restricted to low magnetic fields and therefore does not allow us to perform accurate measurements of the Hall effect. To estimate the carrier density induced by the ionic gate, we compare to Refs. 13 and 26, which establish a relationship between $T_c$ and $n_n$ that is consistent across many devices. Based on $T_c^R$ at $V_{BG} = 0$, device B has a normal carrier density of $1–2 \times 10^{14}\ cm^{-2}$ [26]. Device A has a lower critical temperature, which could be due to under- or over-doping. At $V_{BG} < 0$, $T_c^R$ increases by -0.4 K for a change of $10^{12}\ cm^{-2}$ in the carrier density, which is in agreement with the increase in $T_c$ observed in Refs. 13, 26, 33 on the underdoped side of the dome. We, therefore, assume in the following that device A is underdoped with a carrier density of $0.6–0.8 \times 10^{14}\ cm^{-2}$. Compared to these carrier densities, $n_n$ decreases by at most 17% and 11%, in device A and device B respectively, across the accessible backgate range. However, we observe larger decreases of 25% and 76%, respectively, in the magnitude of $\chi$. Similarly, the normal-state resistance of both devices changes more significantly than can be explained by the carrier density, assuming the values for $n_n$ mentioned above, which are based on comparing to previously reported behavior[13,26]. This suggests that the microscopic disorder in the samples is varying with $V_{BG}$. From $R_\square$, we estimate the mobility at $V_{BG} = 0$ to be $300\ cm^2/V\ s$ in device A and $40\ cm^2/V\ s$ in device B assuming $0.7 \times 10^{14}\ cm^{-2}$ and $1.5 \times 10^{14}\ cm^{-2}$ for the carrier densities respectively. There is a significant degree of variation in the mobility and normal-state resistance of $MoS_2$ devices reported in the literature[13,21,28]. However, our devices fall within the observed range, and the data suggest that device B has higher microscopic disorder than device A based on its

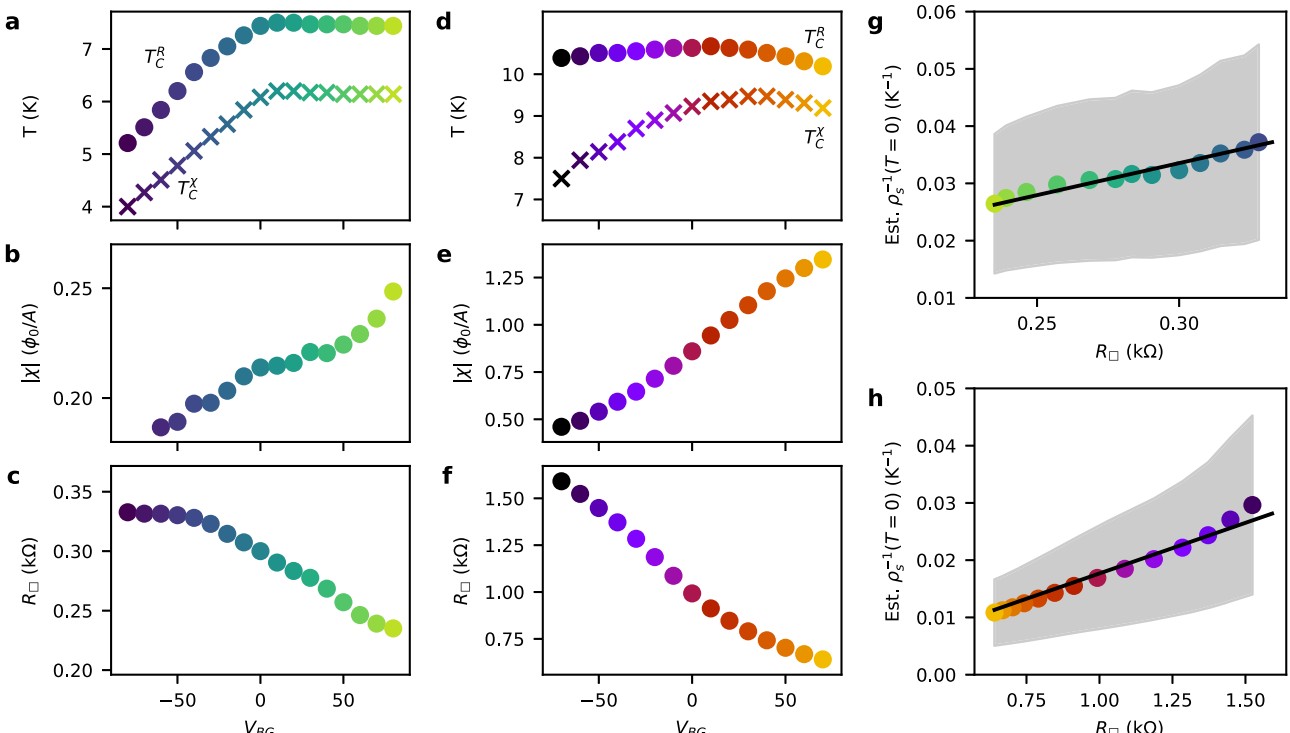

**Fig. 3 | Correlation between device resistivity and superfluid stiffness.**
**a**–**c** Summary of properties of the superconducting state in device A versus $V_{BG}$.
**a** Two measures of the critical temperature: $T_c^R$ at which the resistance has
decreased by 10% from the normal value and $T_c^\chi$ at which the superfluid response
starts to exceed our noise floor. **b** $\chi$ extracted at $T = 0.9 T_c^\chi$. **c** $R_\square$ at 8 K. **d**–**f** Same as
**a**–**c** but for device B. **e** $\chi$ is shown at $T = 0.55 T_c^\chi$, for **f** $R_\square$ is shown at 12 K. **g** Inverse of

the superfluid stiffness at zero temperature versus $R_\square$. See the main text for the
method of estimation of $\rho_s(T = 0)$. The colors of the circles match the colors in **a**–**c**.
The uncertainty in the inverse superfluid stiffness from uncertainty in the height of
the SQUID is indicated by the gray band. Fit to Eq. (1) shown in black. **h** Same as
**g** but for device B.

mobility. This may seem surprising given that device B has a larger
residual resistance than device A. However, as we discuss in Sup-
plementary Discussion 2, we believe the residual resistance arises
from large-scale non-uniformities rather than microscopic electro-
nic disorder.

Disorder in 2D superconductors can strongly reduce the super-
fluid density compared to the normal-state carrier density even at
$T = 0$. In the dirty limit, in which the elastic scattering rate $1/\tau$ exceeds
the superconducting gap, $\Delta$, the fraction of carriers forming the
superconducting condensate is expected to be $n_s(T = 0)/n_n \approx 2\Delta/(\hbar/\tau)$[6],
where $n_s(T = 0)$ is the superfluid density at zero temperature. Using
$R_\square = m^*/(n_n e^2 \tau)$, we can relate the superfluid stiffness and the sheet
resistance through

$$\rho_s(T = 0) \approx \frac{\Delta \hbar}{2 k_B e^2 R_\square}. \qquad (1)$$

To compare this relationship to our data, we model $M_{geo}$ to
convert $\chi$ into $\rho_s$. $M_{geo}$ depends on our measurement geometry, such
as the height and dimensions of the SQUID and the dimensions of
the sample (see Supplementary Methods 1). We then estimate
$\rho_s(T = 0)$ by fitting a phenomenological BCS model to the tempera-
ture dependence of $\rho_s$[1] as discussed in more detail below (also see
Supplementary Discussion 3). We constrain the fits to below a fixed
fraction of $T_c^\chi$. Due to the limited temperature range, we cannot
perform this analysis for traces corresponding to the most negative
$V_{BG}$ values from both devices. In Fig. 3g, h we plot $1/\rho_s(T = 0)$ versus
$R_\square$ for device A and B, respectively. The shaded gray areas reflect
several systematic uncertainties in estimating $M_{geo}$ which do not
affect the relative changes of $\rho_s$ with backgate or temperature. The
fit to Eq. (1) is shown in black and we extract $\Delta = 0.4 \pm 0.2$ meV and

$\Delta = 2.6 \pm 1.2$ meV for device A and B, respectively. For device B, this is
in agreement with $\Delta = 1.75$ meV extracted from tunneling
spectroscopy[26]. Tunneling has not been performed in underdoped
devices; however, $\Delta$ for device A is similar to the gap of overdoped
devices of comparable $T_c$. Combined, our measurements suggest
that the backgate modifies the superfluid stiffness via both disorder
and carrier density, rather than density alone. In some devices, such
as device B, disorder tuning is the predominant mechanism. Fur-
ther, assuming the effective mass $m^*$ to be equal to the free electron
mass, we estimate the zero temperature superfluid density at
$V_{BG} = 0$ based on the measured superfluid stiffness. We find
$1.4 \pm 0.6 \times 10^{13}$ cm$^{-2}$ for device A and $2.7 \pm 1.4 \times 10^{13}$ cm$^{-2}$ for device B.
These values are substantially lower than the inferred normal-state
density, suggesting that disorder substantially reduces the super-
fluid density.

Potential sources of electrostatic disorder in our devices are
the SiO$_2$ substrate, the ionic gate, and intrinsic disorder in the MoS$_2$
flake. As a function of backgate voltage, the carrier density,
screening properties, and the shape of the confinement potential
may change. Refs. 33, 34 study few-layer MoS$_2$ devices dual-gated
by an ionic gate and a backgate similar to ours. They report sig-
natures of a low-density metallic layer forming at the bottom of the
MoS$_2$ flake only at positive backgate voltage, whereas at negative
backgate voltage the carrier density of the top layer decreases.
Such a metallic layer could modify the electron mean free path in
the top layer by screening the disorder from the SiO$_2$ substrate,
which is likely significant given the reported enhancement in
mobility of gated MoS$_2$ devices that are placed on hBN[35]. At nega-
tive backgate voltage, the screening length in the top layer may
increase with negative backgate voltage, increasing the effective
disorder. In our data, the critical temperature in sample A and the

                                                                    

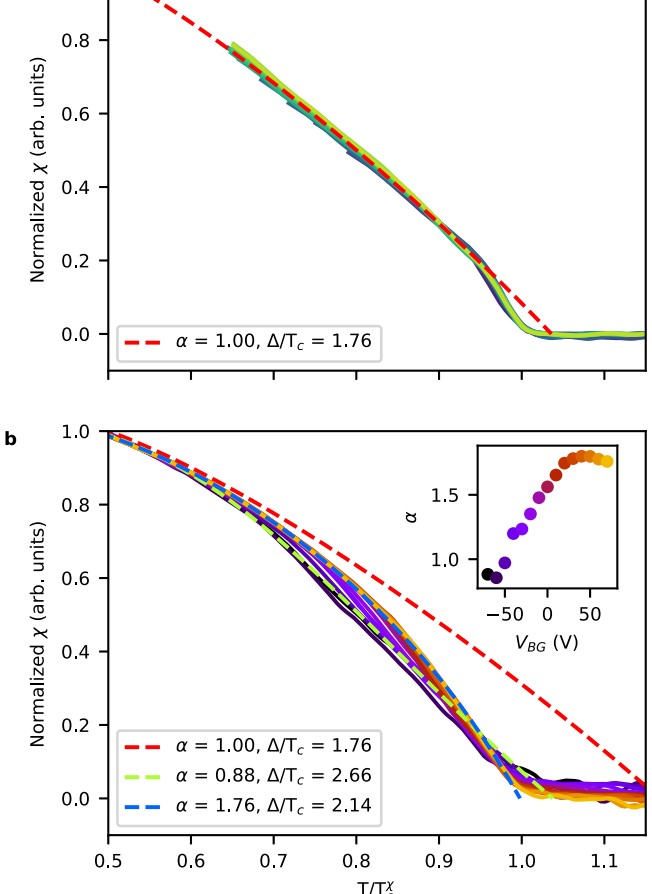

**Fig. 4 | Normalized superfluid response curves. a** Gate-tuned superfluid response $\chi$ from device A with the temperature axis scaled by $T_c^\chi$ and the vertical axis by $\chi$ at 0.9 $T_c^\chi$. Dashed red curve shows a fit to $\chi$ below 0.9 $T_c^\chi$ to the phenomenological model in Eq. (2) assuming a BCS dependence with $\Delta/T_c^{BCS} = 1.76$ and a shape parameter $\alpha = 1$. **b** Gate-tuned superfluid response of device B with the axes scaled similarly as in **a**, but using $\chi$ at 0.55 $T_c^\chi$. Red dashed curve shows same fit as in **a** to data below 0.55 $T_c^\chi$. The green and blue dashed curves show fits of Eq. (2) to the full temperature range at $V_{BG} = -70$ V and $V_{BG} = 70$ V, respectively, with no constraints imposed on $\Delta/T_c$ and $\alpha$. The inset shows the evolution of the shape parameter, $\alpha$, obtained from fits at all values of $V_{BG}$. Error bars are smaller than the markers.

finite resistance in the superconducting state in both devices (see Figs. 3a, d and Supplementary Fig. 7) show different behavior at positive and negative backgate voltages. However, the superfluid response and normal-state resistance evolve relatively continuously as a function of backgate voltage. If the mechanism of disorder tuning changed at $V_{BG} = 0$, one might expect some asymmetry in these parameters as well, which is not observed. Therefore, the role of the proposed mechanisms in tuning disorder in our dual-gated devices, and whether any other effects are important, remains an open question.

## Signatures of the BKT transition
Next, we discuss the magnetic response close to the superconducting transition. Device A and B show significantly different behavior. In device A (Fig. 2b), we observe a sharp rise in the superfluid response near $T_c^\chi$, followed by a change to a lower slope giving rise to a kink in the curves. In device B (Fig. 2d), we observe a smooth rise in the superfluid response throughout the accessible temperature range. A jump in the superfluid response is expected in

a 2D superconductor close to $T_c$ due to the BKT transition[7,8]. During the resistive transition, weak fluctuation diamagnetism may exist, however, we expect the corresponding signal size to be below our sensitivity[8]. Similarly, we are not sensitive to weak diamagnetism emerging from small superconducting regions. Just below the transition, the superfluid stiffness is predicted to satisfy the universal condition $\rho_s/T_{BKT} = 2/\pi$. In Fig. 2b, d, we include a line corresponding to $2T/\pi$ converted to $\chi$ in black and the systematic error from the uncertainty in $M_{geo}$ in gray. For device A, the sharp rise in $\chi$ near $T_c$ ends at the lower edge of the universal condition error band. Instead of a jump, there is a kink, which is consistent with a BKT transition broadened and modified by a combination of finite-size and disorder effects[36]. In device B, however, we do not observe any features that match the expected form of a BKT transition.

To highlight how the temperature dependence of $\chi$ evolves as a function of $V_{BG}$, we normalize the superfluid response versus temperature curves in Fig. 4. The vertical axis is scaled by the superfluid response at 0.9 and 0.55 $T_c^\chi$ for device A and B, respectively. The horizontal axis is scaled by $T_c^\chi$. For device A, the curves collapse. Conversely, in device B, the curves differ in the range of 0.7 to 1.0 $T_c^\chi$.

We compare the normalized curves to a phenomenological model for s-wave superconductors[1]:

$$\chi = \chi_0 \left(1 - \frac{1}{2T} \int_0^\infty \cosh^{-2}\left(\frac{\sqrt{\epsilon^2 + \Delta^2}}{2T}\right) d\epsilon \right). \qquad (2)$$

Here $\chi_0$ is the zero temperature response, and $\Delta$ is temperature-dependent gap given by:

$$\Delta(T) = \Delta_0 \tanh\left(\frac{\pi T_c^{BCS}}{\Delta_0}\sqrt{\alpha\left(\frac{T_c^{BCS}}{T} - 1\right)}\right), \qquad (3)$$

where $\alpha$ is the shape parameter governing the opening of the gap, $T_c^{BCS}$ is the critical temperature, and $\Delta_0$ is the size of the gap at zero temperature. We first constrain $\Delta_0/T_c^{BCS} = 1.76$ and $\alpha = 1$ as expected for a BCS superconductor, leaving only two free parameters, $T_c^{BCS}$ and $\chi_0$. We fit to data below .9 $T_c^\chi$ for device A and .55 $T_c^\chi$ for device B, because the superfluid response is in clear disagreement with the phenomenological model above the cutoff temperature for device A, and in device B, the shape of the onset is changing with $V_{BG}$. The resulting fits for the highest backgate voltages are shown as red dashed lines in Fig. 4a, b. In device A, $T_c^{BCS}$ is slightly above the onset of susceptibility consistent with a small temperature range above $T_c^\chi$ in which the superfluid response is suppressed due to a BKT transition, which is not captured in the phenomenological model. The fitted value of $T_c^{BCS}$ is sensitive to details of the fitting such as an initial guess and the exact temperature range. However, the extracted low-temperature value $\chi_0$ is comparably robust and was used to estimate $\rho_s(T=0)$ for Fig. 2g, h (see Supplementary Discussion 3). The onset of diamagnetism in both devices is not captured by the simple model, which could be due to a number of reasons. In particular, disorder can modify the onset of diamagnetism even in 3D superconductors. Within the phenomenological model in Eq. (2), disorder causes an increase in the shape parameter[1,37]. To explore this further, we fit our data across the entire temperature range without constraints on $\alpha$ and $\Delta_0/T_c$. For device A, we cannot obtain a good fit of the model near $T_c^\chi$, indicating that the kink in the curves is not due to a disorder-modified BCS transition. In device B, good agreement between the model and the data can be achieved for all backgate voltages. The fits for the highest and lowest backgate voltages are shown as dashed blue and green lines, respectively, in Fig. 4b (all fits are shown in Supplementary Fig. 8). The fitted $\Delta_0/T_c$ significantly exceeds the BCS result, and $\alpha$ increases from less than 1 to almost 2 with increasing $V_{BG}$ (see inset to Fig. 4b).

                                                                                    

This dependence is contrary to expectation, as the model suggests that $\alpha$ should decrease with decreasing disorder and hence with increasing $V_{BG}$, because the change of $R_\square$ with $V_{BG}$ suggests lower disorder at more positive $V_{BG}$.

For device A, the deviation of the superfluid response from Eq. (2) is likely caused by phase fluctuations. That is, the kink in $\chi(T)$ results from a BKT jump in $\rho_s(T)$ slightly broadened by the interplay of finite-size and disorder effects. Although device B does not show a similarly clear feature, it is likely that similar effects are at play. For a given system size, within the disorder-modified BKT paradigm we expect that stronger disorder (i.e., decreasing $V_{BG}$) will cause the superfluid response to become more shallow close to $T_c$[38], which is what we observe. This behavior is also consistent with the generally stronger disorder in device B compared to device A as indicated by the normal-state carrier mobility. Therefore, our data suggest that $MoS_2$ represents a crossover system, where the superfluid stiffness near $T_c$ is governed by phase fluctuations, but a clear signature of a BKT transition may or may not be present depending on doping and other parameters.

## Discussion

In conclusion, we report the first characterization of the superfluid response of an atomically thin van der Waals superconductor using a local probe that provides sufficient sensitivity to the small sample volume typical in this material family. We find that the superfluid stiffness monotonically increases at low temperatures as the backgate is tuned, even when the critical temperature decreases. Our analysis suggests that our devices are in the dirty limit of superconductivity in which the superfluid stiffness responds to changes in device resistivity. This demonstrates that disorder plays an important role even in crystalline 2D superconductors. Further, we observe direct signatures of a BKT transition in one device, whereas, in another, the universal jump is replaced by a broad region of suppressed superfluid response close to $T_c$. This demonstrates that a clear BKT transition is not ubiquitous in these systems, but can be substantially obscured by disorder. In the present work, our 4 K base temperature prevented characterizing $\chi$ at a small fraction of $T_c$. Future work extending to lower temperatures will be sensitive to the presence of nodes in the superconducting gap, which would be a sign of an unconventional order parameter[26,39–41].

## Methods
### Device fabrication
We exfoliated bulk $MoS_2$ crystals (HQ Graphene) onto flexible polydimethylsiloxane (PDMS) substrates (Dow Sylgard 184), and identified 3–10 layer flakes by optical contrast. Using a polycarbonate film supported by a PDMS stamp, flakes were individually transferred onto pre-patterned $SiO_2/Si$ substrates. The polycarbonate film was then stripped using chloroform. The silicon substrate is highly doped to both thermalize the sample and act as a backgate. The pre-patterned substrates are nearly entirely covered with evaporated platinum used as the ionic gate electrode except for a clear area for the device in the center, and a thin strip in which connections run from the device to electrical bond pads. To keep leakage between this large platinum pad and the backgate low, we use Si wafers with high-quality dry chlorinated oxide (Nova Wafers). After transfer of the $MoS_2$ flake, Ti/Au contacts are fabricated by a bilayer PMMA/MMA electron beam lithography lift-off process. In all e-beam steps, cold 3:1 IPA:DI developer is used. To prevent delamination of the metal, we used a mild remote oxygen plasma to descum the contact areas before deposition of the metal. To minimize contact resistance, only 1 nm of Ti is used for adhesion, achieving a contact resistance on the order of 5 kOhms-μm with the device doped into the superconducting state. After lift-off in acetone, we used electron beam lithography to pattern a single-layer PMMA etch mask. Using a dry $CHF_3/O_2$ etch, the $MoS_2$ was etched into

the final device shape. The device is then baked at 250 °C for 4 hours in UHV (<1e-6 Torr), to remove any absorbed water. An ionic gel is prepared by mixing 1% polystyrene-poly(methyl methacrylate)-polystyrene triblock polymer (Polymer Source), 10% diethylmethyl(2-methoxyethyl)ammonium bis(trifluoromethylsulfonyl)imide (Sigma-Aldrich), and balance ethyl propionate. Ethyl propionate was chosen due to its low volatility, ability to dissolve the polymer, and miscibility with the ionic liquid. The ingredients are mixed in a dry nitrogen atmosphere in an all-glass container, and agitated for 24 hrs. Then, under a dry nitrogen atmosphere, we spin-coat the ionic gel at 2000 RPM onto the sample substrate. Without exposure to air, the solvent is removed under vacuum at room temperature for 24 hrs. After this cure, the ionic gel is manually removed over the bond pads and at the edge of the chip, preventing electrochemical reactions with the aluminum bond wire. The completed dual-gated $MoS_2$ device is shown schematically in Supplementary Fig. 1a and optical images of all devices are shown in Supplementary Fig. 1b. Upon loading the sample into the cryostat some exposure to air is inevitable, so the sample is allowed to dehydrate at room temperature in high vacuum in the system for 24 hours prior to cool down. Prior to this dehydration, all connections to the device are kept grounded. After this step, the cryocooler is turned on and allowed to cool to 4 K over 16 hours. When the sample temperature reaches 220 K, a DC voltage is applied between the ionic gate electrode and the device, causing negative ions to accumulate on the ionic gate electrode and positive ions to accumulate on the device. A compensating electronic charge appears in the $MoS_2$ device, inducing on the order of $10^{14}$ cm$^{-2}$ charge carriers[13]. We find that typically a voltage of -5.5 V induces superconductivity, although this is likely sensitive to choices made in fabrication and design. We used 5.6 V, 5.5 V, and 5.5 V applied to the ionic gate electrode for devices A, B, and C, respectively.

### Experimental setup
In our scanning probe microscope, we use commercial piezo stages (attocube) for coarse positioning and custom piezoelectric benders for fine positioning of the SQUID. The sample is anchored to a thermally isolated copper mount, along with a heater and a thermometer. This enables the system, and SQUID, to remain at 4 K while the sample temperature is varied. The on-chip sample leads are thermalized to the copper mount through the large area of the bond pads (60,000 μm$^2$) and the high thermal conductivity of $SiO_2$. The sample wires are thermally anchored at the baseplate of the cryostat (Montana Instruments) using bobbins secured with stycast, and filtered using a QDevil RC filter. At the room temperature breakout box, unused device leads are capped using non-shorting BNC caps. Four-point measurements were acquired by flowing a current of 64 nA in device A and 571 nA in device B. The exact four-point geometry varied between devices, as shown in Supplementary Fig. 1b. We used COMSOL Multiphysics to relate the measured resistance ($R_{4pt}$) to the sheet resistance we report ($R_\square$), finding $R_\square/R_{4pnt}$ = 3.6, 3.6, and 2.9 for device A, B, and C, respectively. All voltage measurements are taken with a Stanford Research SR560 preamplifier with a 100 MOhm input impedance. For the SQUID measurements, the field coil was driven at approximately 1 kHz, and the field coil current was set low enough that the sample response did not vary with the current (250 μA$_{RMS}$ for device A, 360 μA$_{RMS}$ for device B). The SQUID is gradiometric and therefore insensitive to uniform background fields. Further, the field coil is counter-wound around the two pickup loops, minimizing the mutual inductance between the SQUID and field coil when no sample is near the front pickup loop. The signal from the SQUID is amplified by a cryogenic SQUID array amplifier, and room temperature feedback electronics keep the flux in the SQUID fixed by changing the current flowing through on-chip modulation coils. This feedback current is proportional to the flux through the SQUID, and we demodulate it using a lock-in amplifier at the frequency of the field coil drive. An air

core, resistive magnet surrounds the cryostat and was used to zero the out-of-plane background field in all the measurements reported in the main text.

## Data acquisition

We found that a common failure mode of the experiments was delamination of the ionic liquid at low temperatures causing catastrophic damage to the device. These events were often correlated with temperature changes at a finite backgate voltage. Therefore, we adopted the following procedure for our measurement: slow warming and cooling rates of <0.5 K/minute when the SQUID was approached to the sample, and the backgate was kept at zero when the temperature changed. Therefore for the backgate-dependent measurements, the data were acquired with temperature as the outermost sweep: for each temperature, the sample is heated to the chosen temperature, then the backgate is swept, then returned to zero, and then the temperature is changed to the next value. This is the origin of the repeated noise features line to line in some data sets.

The SQUID signal exhibits a small non-zero phase with respect to the field coil drive even far away from a sample. This is likely due to a small parasitic impedance in the electronics and wiring. This effect is negligible on the in-phase response from a superconducting sample. However, its effect on the out-of-phase response can be substantial, because that part is small or zero. Therefore, we characterized the phase shift with the SQUID far away from the sample at all frequencies and rotated the collected data to correct this effect. After this rotation, no out-of-phase response was observed in the data reported in this work.

Two background signals contribute to our measurement signal which we subtract from the raw data. First, the field coil is counterwound around the gradiometric squid to minimize the mutual inductance between the SQUID and the field coil. However, a slight remaining mutual inductance caused by lithographic imperfections is present such that we detect a non-zero offset signal even far above the sample. Second, close to the sample surface but away from the device, we detected a moderately temperature-dependent magnetic response likely originating from the ionic liquid. Therefore, for all measurements, we collected a temperature-dependent background signal nearby the superconducting disk at the same height. These background data are plotted in Supplementary Fig. 2 and include the offset in mutual inductance. We subtract this background from the raw magnetic response data. After that, a small amount of smoothing was applied to each resulting $\chi(T)$ curve. The data was sampled more finely in temperature than needed given the width of observed features, and therefore the smoothing only reduces the noise on the traces without changing any features. A three-pass local regression algorithm was used for smoothing, with 5% of the total data used to smooth at each point.

## Data availability

The raw data used in the preparation of the figures are available at https://zenodo.org/record/7647326.

## Code availability

The Python notebooks used to produce the figures and relate the magnetic response to the Pearl length are available at https://zenodo.org/record/7647326.

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

## Acknowledgements

We thank Eric Smith for advice on cryogenic instrumentation and Deb-deep Jena, Huili Xing, Guen Prawiroatmodjo, Eun-Ah Kim, Yi-Ting Hsu, and Debanjan Chowdhury for fruitful discussions. This work was pri-marily supported by the National Science Foundation under Grant No. DMR-2004864, and partially supported by the Cornell Center for Materials Research with funding from the NSF MRSEC program (DMR-1719875). In addition, N.T. acknowledges support from NSF-DMR 2138905. This work was performed in part at the Cornell NanoScale Facility, an NNCI member supported by NSF Grant NNCI-2025233.

## Author contributions

A.J. and K.C.N. designed the experiments. A.J., B.T.S., and G.M.F. built the instrument. A.J. fabricated the devices, performed the measure-ments, processed the data, and performed modeling to estimate the SQUID height and relate stiffness and magnetic response. M.L. advised implementing liquid gating and cryogenic measurements. A.J., Y.L.L., N.T., and K.C.N. analyzed the data. A.J. and K.C.N. wrote the manuscript with input from all authors.

## Competing interests

The authors declare no competing interests.
