## [Peer Review File · Nature Communications]

Superfluid response of an atomically thin gate-tuned van der Waals superconductorREVIEWER COMMENTS

Reviewer #1 (Remarks to the Author):

The manuscript of Jarjour et al. presents a very interesting study of superfluid response in atomically thin flakes of superconducting MoS₂. Utilizing a microscopic scanning SQUID, the authors performed a detailed study of the magnetic susceptibility in patterned MoS₂ flakes vs. temperature and backgate voltage and compare them to simultaneous transport measurements. Even though the origin of several of the observed features, including finite residual resistance and nonmonotonic critical temperature is unclear, the experimental data are valid and the analysis is extensive.

The main impact of the paper is the first measurement of superfluid response in a van der Waals superconductor flake which opens the prospect of spatially resolved studies of superconducting properties in a wide range of exotic superconducting states in moiré materials. This work will set the benchmark for such future experiments and I therefore recommend its publication in Nature Communications.

In order to facilitate future studies and evaluate their feasibility, it would be very useful to add some additional information and analysis in the manuscript, if possible:

1. Extract the superfluid density within the experimental uncertainty.
2. Specify the estimated average applied ac magnetic field excitation.
3. Translate the measured superfluid response into the induced ac current density and compare it to the critical current density measured in transport or available in the literature.
4. From the above results, estimate the kinetic inductance and compare it to the evaluated geometric inductance of the disk-shaped flake. Alternatively, evaluate the measured ac screening relative to the ideal Meissner screening for the given experimental setup geometry.

Reviewer #2 (Remarks to the Author):

The manuscript entitled "Superfluid response of an atomically thin, gate-tuned van der Waals superconductor" by Jarjour et al. reported original research on diamagnetism in a TMD-based 2D superconductor, MoS₂. The study went beyond the widely used electrical transport. This additional probe showed that back gate modification of the carriers induced by ionic gating strongly depends on uniformity. And the superfluid response deviates from simple BCS-like behavior in some samples. This research represents an alternative direction in characterizing the 2D superconductivities induced by ionic gating. Therefore, I highly recommend the publication in Nature Communications.

I would like to add the following points.

1. For the device configuration, the ionic channel is 2 μm in depth. Therefore, the gating is expected to differ from the bulk ionic material used in previous reports. Namely, the ionic motion is expected to be more restricted than bulk ionic media. I need a comment in the manuscript about whether it might cause the strong non-uniformity observed in the experiments.
2. The authors mentioned the limit of weak screening, which is essential for the simplified picture and analysis used in the manuscript. Therefore, we need a better justification of the validity.
3. Since the resistivity measurement in the transport tests a global sample response, or more precisely, the voltage response from the non-superconducting path between the voltage probes, the path might not entirely match the position of SQUID measurement, namely at a fixed position close to the center. If the sample is not uniform in finer grain, the analysis in Fig. S5 might not be able to catch it. Authors are expected to comment on this.

4. In Fig. 2, besides the significant difference between panels a and c, in quite contrasting gate responses, the $R(T)$ dependence of samples A and B both show, with VBG depletion, finite resistances, remaining down to base temperatures. Therefore, it is fair to add that even though panel c shows VBG response indicating differences in uniformity - sample B appears better than sample A with a higher T_c and a lower residual resistance - the superconducting states are still different from that reported in the literature, for example, in Ref. 13.

5. It is interesting to observe that the contrasts shown in Fig. 2 are not causing fundamental differences in the physical parameters extracted from the analysis. In Fig. 3, the gate dependence of parameters shows similar trends for samples A and B. The difference is only in magnitude, especially in comparing panels b and e. Nevertheless, the behaviors are quite different in the comparisons in Fig. 4. To me, the conclusion of stronger non-uniformity in sample B is inconsistent with the comparison shown in Fig. 2.

A few more points

6. In the abstract, "As a consequence, characterization of the diamagnetic response of the superfluid to an applied magnetic field, a defining property of any superconductor, has been lacking."

This statement is not entirely correct. Diamagnetism has been studied in mutual inductance settings using conventional coils for ionic gating (doi:10.1038/nature09998). Therefore, the claim should be modified slightly, to be precise.

7. For all places citing Supplementary Information, the readers will benefit from more specific references, say to which session.

Reviewer #3 (Remarks to the Author):

Alexander Jarjour et al. perform simultaneous scanning SQUID and transport measurements on few-layer MoS₂ superconductors that are produced by placing the thin film in contact with an ionic gel that heavily dopes the film. Additional tuning of the properties is achieved by using an electrostatic back gate electrode. The authors are able to see a signature of the magnetic susceptibility of the superconductor in the SQUID response even though it is several microns from the sample surface, which is buried under the ionic gel. They are able to simultaneously measure the longitudinal resistance and measure the resistive transition as well. Both the transitions are broad (~ 1.5 K), and the authors analyze the susceptibility data to understand whether the physics of the Berezinskii-Kosterlitz-Thouless (BKT) transition is at play in the observed resistance and susceptibility. The two different samples measured give different answers to this question.

In general, the ability to perform measurements other than simple resistivity is an exciting new dimension to the study of ultrathin superconductors. The fact that the authors have achieved this through an ionic gel for MoS₂ is very impressive, and so I feel that the work definitely deserves to be published. The figures and writing are generally high quality. I recommend publication after the authors address the following questions.

The authors say that the superconductivity is confined to the top layer, and this is a reasonable statement based on the previous work. At the same time, the authors assume that the back gate voltage is tuning the density of the superconducting layer. These two assumptions seem incompatible with each other. Since the sample is 3-10 layers thick, by the same logic used for the superconducting layer, all of the doping produced by the back gate should affect the bottom layer only, which is not superconducting. It is difficult to see how the back gate does anything to affect the superconductivity. Even if the back gate charges traps in the SiO₂, by this logic the bottom layer should screen these charges and the superconducting layer should be unaffected. So the authors need to make a more consistent picture for what the back gate is doing. Is it possible that the back gate somehow draws ions from the ionic gate? This would also be consistent with the

anomalously large effect of the back gate observed.

In nearly all cases, the resistive superconducting transitions are nearly completed before the susceptibility sees the superconductivity. The authors do not really address this point. If this is because there are special paths that dominate the transport, is it then reasonable to use the sheet resistance from the normal state in the superfluid density calculations?

Regarding the effect of disorder - the authors invoke extrinsic disorder like the SiO₂ substrate. In my opinion, a much simpler effect is the change in the potential landscape as a function of chemical potential. It is well known that the TMD semiconductors show poor transport properties near the band edge. The authors could comment on this possibility and perhaps include appropriate references.

Thank you for your detailed reading and thoughtful comments on our manuscript "Superfluid response of an atomically thin, gate-tuned van der Waals superconductor". We provide a point-by-point response to all comments below. Descriptions of revisions to the manuscript are shown in bold.

Reviewer #1:

The manuscript of Jarjour et al. presents a very interesting study of superfluid response in atomically thin flakes of superconducting MoS₂. Utilizing a microscopic scanning SQUID, the authors performed a detailed study of the magnetic susceptibility in patterned MoS₂ flakes vs. temperature and backgate voltage and compare them to simultaneous transport measurements. Even though the origin of several of the observed features, including finite residual resistance and nonmonotonic critical temperature is unclear, the experimental data are valid and the analysis is extensive.

Thank you for your positive feedback on our manuscript. We are glad to hear that you found our study of superfluid response in atomically thin flakes of superconducting MoS₂ interesting. We are grateful for your recognition of the validity of our experimental data and the thoroughness of our analysis. As you noted, there are still several unexplained features in our data, and we hope that our study will inspire further research into these phenomena.

The main impact of the paper is the first measurement of superfluid response in a van der Waals superconductor flake which opens the prospect of spatially resolved studies of superconducting properties in a wide range of exotic superconducting states in moiré materials. This work will set the benchmark for such future experiments and I therefore recommend its publication in Nature Communications.

Thank you for your kind words and explaining the impact of our work. We indeed hope that our study will inspire similar measurements in other vdWs superconductors including moiré materials, and we look forward to seeing the exciting results that may arise from such studies. We are delighted to hear that you recommend our manuscript for publication in Nature Communications.

In order to facilitate future studies and evaluate their feasibility, it would be very useful to add some additional information and analysis in the manuscript, if possible:

1. Extract the superfluid density within the experimental uncertainty.

Thank you for bringing this to our attention. **We have added an estimate of the superfluid density, given assumptions about the effective mass, to the manuscript.**

2. Specify the estimated average applied ac magnetic field excitation.

Thank you for suggesting this. We agree that this should be clarified in the manuscript. **We have added an estimate of the AC magnetic field excitation to the main text.** We note that we choose the excitation such that the response of the sample is in the linear regime. We check this by varying the excitation and verifying that the response scales linear with the excitation. **In the revised manuscript, we explain this aspect more carefully.**

3. Translate the measured superfluid response into the induced ac current density and compare it to the critical current density measured in transport or available in the literature.

We agree that this is a useful comparison. **We have added a paragraph to the supplemental materials to address this point, and briefly mention it in the main text together with the estimate for the applied excitation (see point 2.).**

4. From the above results, estimate the kinetic inductance and compare it to the evaluated geometric inductance of the disk-shaped flake. Alternatively, evaluate the measured ac screening relative to the ideal Meissner screening for the given experimental setup geometry.

Thank you for requesting this analysis. **We have added a section to the supplement showing that our samples are in the weak screening limit**, i.e. the ac screening we measure is much weaker than the ideal Meissner screening. This analysis is done by simulating the expected ac response as a function of the Pearl length of the sample. From this we find that the Pearl length is much larger than the device size for all signal levels that we observe. The simulation also includes the signal we would observe in the ideal Meissner screening limit. **In the main text, we clarify this explicitly and reference the supplement for details.**

Reviewer #2:

The manuscript entitled “Superfluid response of an atomically thin, gate-tuned van der Waals superconductor” by Jarjour et al. reported original research on diamagnetism in a TMD-based 2D superconductor, MoS₂. The study went beyond the widely used electrical transport. This additional probe showed that back gate modification of the carriers induced by ionic gating strongly depends on uniformity. And the superfluid response deviates from simple BCS-like behavior in some samples. This research represents an alternative direction in characterizing the 2D superconductivities induced by ionic gating. Therefore, I highly recommend the publication in *Nature Communications*.

Thank you for carefully reading our manuscript and for your positive assessment. We are pleased that you appreciate the value of our study of the superfluid response as an additional way to characterize 2D ionic gated superconductors. We greatly appreciate the recommendation of publishing our manuscript in *Nature Communications*.

I would like to add the following points.

1. For the device configuration, the ionic channel is 2 μm in depth. Therefore, the gating is expected to differ from the bulk ionic material used in previous reports. Namely, the ionic motion is expected to be more restricted than bulk ionic media. I need a comment in the manuscript about whether it might cause the strong non-uniformity observed in the experiments.

Thank you for this insightful comment regarding the ionic gate. We agree that the geometry of our ionic gate is different from that used in previous work on vdWs superconductors, and this could in principle have an effect on the behavior of the ionic gate. As we discuss more in a subsequent response, two primary differences exist between our data and that of Ref. [13] in the manuscript (Ref. [1] in this letter): our devices exhibit a higher normal state resistance and a finite resistance in the superconducting state. Certainly, the behavior of the ionic gate might play a role in these differences. However, both of these phenomena are observed in traditionally liquid gated devices, with sheet resistances as high as 1.6 k Ω/\square being observed (see Ref. [2] of this letter). Ionic gates, in general, exhibit a substantial degree of variability due to sample preparation as well as the exact method of setting the gate (our preparation and procedures are described in detail in the Methods section). Therefore, it is difficult to assess whether the gel plays a significant role. We note that our gel is approximately 90% ionic liquid by mass before it is spin coated onto the sample, and even higher after the solvent is removed post-spin coat.

To address this question, we make a few changes in the manuscript:

- 1. When we describe our devices we make explicitly clear that our gating geometry is different from those previously reported in the literature.**
- 2. When we discuss the finite resistance (large-scale disorder) we point out that our non-traditional gate may play a role, but that previously reported traditionally gated devices have also exhibited this effect.**
- 3. When we discuss the inferred mobility of our devices, we mention that a substantial degree of variability exists between the mobilities of devices previously reported in the literature.**

2. The authors mentioned the limit of weak screening, which is essential for the simplified picture and analysis used in the manuscript. Therefore, we need a better justification of the validity.

Thank you for pointing out that this needs clarification. As mentioned in our response to comment 4 by Reviewer #1, **we have added a section to the supplement to show that our measurements are in the limit of weak screening. In the revised manuscript, we refer to this section at relevant places.**

3. Since the resistivity measurement in the transport tests a global sample response, or more precisely, the voltage response from the non-superconducting path between the voltage probes, the path might not entirely match the position of SQUID measurement, namely at a fixed position close to the center. If the sample is not uniform in finer grain, the analysis in Fig. S5 might not be able to catch it. Authors are expected to comment on this.

Thank you for the opportunity to clarify our reasoning to include the analysis in former Fig. S5 (now Supplementary Figure 6) in the manuscript. To start, we note that Fig. S5 and the associated discussion are meant to give a toy model that can explain the finite resistance observed in the superconducting state and that is compatible with the superfluid response images that we took of our devices. The reason we include this scenario is to emphasize that there are reasonable explanations for the finite resistance that do not require the resistance to arise from the superconducting state itself.

An extended non-superconducting line defect or region traversing the sample might also explain the observed finite resistance. For example, consider a device with contacts 1-4 in order along the circumference of the device. In a measurement in which current is sourced from contact 1 to 2 and voltage is probed between contact 3 and 4, a continuous non-superconducting line defect that does not support a supercurrent and that separates contacts 1 and 3 from contacts 2 and 4 would cause a finite resistance. However, because the line defect does not support a supercurrent across it, it would give rise to clear signatures in our spatial maps of the superfluid response. This is true even if the defect was much narrower than our spatial resolution because it would significantly disturb the structure of the supercurrents that flow in response to the excitation from the field coil (see Ref. [3] of this letter). **We have added a section to the supplement showing additional superfluid response images of the devices and discuss how we conclude that there are no non-superconducting line defects present.**

Conversely, one could imagine that the resistive transition instead is dominated by a special superconducting path, this scenario is discussed in our answer to Reviewer 3 below.

Finally, we note that the SQUID is a few micrometers above the sample and the field coil is of comparable size to the sample. Therefore, the magnetic response measurement probes a substantial volume fraction of the entire device, given that the device is approximately 15 micrometers in diameter.

4. In Fig. 2, besides the significant difference between panels a and c, in quite contrasting gate responses, the $R(T)$ dependence of samples A and B both show, with VBG depletion, finite resistances, remaining down to base temperatures. Therefore, it is fair to add that even though panel c shows VBG response indicating differences in uniformity - sample B appears better than sample A with a higher T_c and a lower residual resistance - the superconducting states are still different from that reported in the literature, for example, in Ref. 13.

Thank you for carefully comparing our data on samples A and B. Sample B indeed displays a lower resistance in the superconducting state. However, we nevertheless believe that our data suggests that sample B has a higher level of electronic disorder. We are glad that we have the opportunity to clarify this here and in the revised manuscript.

As discussed in response to the previous comment, we believe that the finite resistance is consistent with micron-scale non-uniformities in the device. If we apply the toy model of a non-superconducting boundary region, we can conclude that the width of the non-superconducting region in sample B would likely be smaller than in sample A. Therefore, sample B could be considered “better” in the sense that this would imply that a larger fraction of the area of this device

becomes superconducting. However, when we state in the manuscript that sample B has a higher level of disorder, we mean to refer to the level of microscopic disorder which causes electron scattering and thereby increases sheet resistance and reduces the superfluid response. Sample B has a higher normal state resistance compared to sample A. Based on the critical temperatures of sample A and sample B and their dependence on gating, we conclude that sample A has a lower carrier density than sample B. To relate these quantities, we assume that the critical temperature of our samples generally follows the density dependence reported in the literature (See Refs. [1,4] of this letter). Under this assumption, sample B has a lower carrier mobility, suggesting it has more microscopic disorder. **In the revised manuscript, we emphasize more strongly that this conclusion is based on assuming that the critical temperature of our samples follow the density dependence reported in the literature.**

Regarding the differences between our samples and those previously reported in the literature, we agree that ionic gated samples often exhibit a substantial degree of variability, even among samples prepared by the same group, and even in some cases the same sample in multiple cooldowns (for example, see Ref. [4] of this letter). Thus, it is important to consider whether our devices fall within the distribution of devices previously reported, or whether they are distinct. Our devices are substantially different from those in Ref. 13 of the manuscript in fabrication, as those are fabricated from approximately 20 nm-thick flakes, whereas ours are fabricated from few-layer flakes, similar to devices reported in Ref. [4] of this letter. Further, there is a wide range of normal state sheet resistances reported in the literature for superconducting MoS₂, ranging from less than 100 Ω/□ in Ref. [1] to 1.6 kΩ/□ in Ref. [2] of this letter. Thus, our sheet resistances are not unusual, although there is a great degree of variability in this parameter, suggesting strong variation in the disorder in these devices. We cannot compare the normal state density as we do not have the ability to perform Hall measurements in our scanning SQUID microscope. Regarding the finite resistance, with the limited data available in the literature, it seems that thicker devices may be less susceptible to this imperfection, as both our devices and those of Ref. [4] of this letter exhibit non-zero superconducting resistance. As we briefly mentioned in the main text, our disk geometry is especially sensitive to a non-superconducting boundary region, a scenario explored in the former Fig. S5 (now Supplementary Figure 6). **To highlight this point, we have added an analysis to the supplement in which we compare the effect of a normal conducting region of the same width and resistivity on the resistivity extracted from a measurement on a Hall bar and a disk-shaped sample. Further, we have clarified in the main text that disorder as measured by the mobility can vary substantially between MoS₂ devices.**

5. It is interesting to observe that the contrasts shown in Fig. 2 are not causing fundamental differences in the physical parameters extracted from the analysis. In Fig. 3, the gate dependence of parameters shows similar trends for samples A and B. The difference is only in magnitude, especially in comparing panels b and e. Nevertheless, the behaviors are quite different in the comparisons in Fig. 4. To me, the conclusion of stronger non-uniformity in sample B is inconsistent with the comparison shown in Fig. 2.

Again, thank you for carefully comparing the behavior of sample A and B. In Fig. 3 b and e, we show the backgate dependence of the superfluid response at a fixed fraction of the critical temperature. For both devices, we observe a monotonic increase with increasing gate voltage that appears uncorrelated with the dependence of the critical temperature on backgate voltage.

In the following, we briefly reiterate how we interpret the dependence of the superfluid response on backgate voltage.

Assuming an effective mass for the carriers, the superfluid response can be directly related to the number of electrons condensed in the superconducting state, or the superfluid density. In the clean limit, all normal carriers condense into the superconducting state. Therefore, in the clean limit, it is expected that the superfluid density and hence the superfluid response increases with increasing carrier density. At first, this might seem a good explanation for the trend in Figs. 3 b and e.

However, the measured superfluid response is overall too weak to originate from the full normal state carrier density (which we estimate as discussed in response to the previous question). In addition and more importantly, it changes more significantly with backgate voltage than would be expected if the backgate only changes carrier density. Therefore we conclude that our samples are in the dirty superconducting limit, a regime which is approximately reached when $\hbar/\tau > \Delta$ where τ is the electron scattering time and Δ is the superconducting gap. In this regime, changes in τ caused by the microscopic electronic disorder are expected to dramatically impact the superfluid response.

Thus, the fact that samples A and B show the same trend in Fig. 3 b and e implies that in both devices τ and likewise the mobility increases with increasing backgate voltage. This conclusion is further supported by the decrease in normal state resistance with increasing backgate voltage.

Fig. 4 highlights differences in the onset of the superfluid response close to the critical temperature in devices A and B. As the reviewer points out, in Fig. 4 we do observe some qualitative difference. Disorder, via various mechanisms, can broaden the jump in superfluid density expected in a BKT transition. Therefore, the broader transition of device B is consistent with higher disorder.

We hope that this summary explains why we believe that stronger microscopic electronic disorder in device B compared to device A is consistent with our data.

In the revised manuscript we explain more clearly why we believe that device B has higher electronic disorder.

A few more points

6. In the abstract, “As a consequence, characterization of the diamagnetic response of the superfluid to an applied magnetic field, a defining property of any superconductor, has been lacking.”

This statement is not entirely correct. Diamagnetism has been studied in mutual inductance settings using conventional coils for ionic gating (doi:10.1038/nature09998). Therefore, the claim should be modified slightly, to be precise.

Thank you for bringing to our attention that this sentence is misleading. We meant to highlight that measurements of diamagnetism in superconducting vdWs devices prepared by mechanical exfoliation are lacking and thought this was clear from the previous sentence. We agree that the diamagnetic of many superconductors has been measured before including ionic gated

superconductors as in the provided reference. **To avoid any misunderstanding, we have added “these materials” also to the sentence in question.**

7. For all places citing Supplementary Information, the readers will benefit from more specific references, say to which session.

Thank you for pointing this out, and we agree that this would improve the readability of our manuscript. **We have implemented this suggestion throughout the revised manuscript.**

Reviewer #3:

Alexander Jarjour et al. perform simultaneous scanning SQUID and transport measurements on few-layer MoS₂ superconductors that are produced by placing the thin film in contact with an ionic gel that heavily dopes the film. Additional tuning of the properties is achieved by using an electrostatic back gate electrode. The authors are able to see a signature of the magnetic susceptibility of the superconductor in the SQUID response even though it is several microns from the sample surface, which is buried under the ionic gel. They are able to simultaneously measure the longitudinal resistance and measure the resistive transition as well. Both the transitions are broad (~1.5 K), and the authors analyze the susceptibility data to understand whether the physics of the Berezinskii-Kosterlitz-Thouless (BKT) transition is at play in the observed resistance and susceptibility. The two different samples measured give different answers to this question.

Thank you for this careful reading and summary of our manuscript.

In general, the ability to perform measurements other than simple resistivity is an exciting new dimension to the study of ultrathin superconductors. The fact that the authors have achieved this through an ionic gel for MoS₂ is very impressive, and so I feel that the work definitely deserves to be published. The figures and writing are generally high quality. I recommend publication after the authors address the following questions.

Thank you for your positive assessment of our manuscript. We greatly appreciate that you describe our work as “very impressive”, and the recommendation to publish it in *Nature Communications* after addressing your comments.

The authors say that the superconductivity is confined to the top layer, and this is a reasonable statement based on the previous work. At the same time, the authors assume that the back gate voltage is tuning the density of the superconducting layer. These two assumptions seem incompatible with each other. Since the sample is 3-10 layers thick, by the same logic used for the superconducting layer, all of the doping produced by the back gate should affect the bottom layer only, which is not superconducting. It is difficult to see how the back gate does anything to affect the superconductivity. Even if the back gate charges traps in the SiO₂, by this logic the bottom layer should screen these charges and the superconducting layer should be unaffected. So the authors need to make a more consistent picture for what the back gate is doing. Is it possible that the back gate somehow draws ions from the ionic gate? This would also be consistent with the anomalously large effect of the back gate observed.

Thank you for the opportunity to clarify our interpretation of the influence of the backgate on the superconductivity. In the manuscript, we cite two electronic transport studies (Refs. [33,34] in the manuscript, Refs. [5,6] in this letter) which suggest that at positive backgate voltages a conducting layer is induced at the bottom of the MoS₂ flake. At negative voltages, no charges are accumulated at the bottom and the backgate changes the carrier density in the top layer. Within this picture, the backgate modifies the carrier density in the top layer only for negative voltages.

In our data, we see some characteristics that show different behavior at positive and negative backgate voltages, most notably the critical temperature in sample A and the finite resistance in the superconducting state in both devices (see Supplementary Figure 7 in the revised manuscript). However, the superfluid response and normal state resistance evolve relatively continuously as a function of backgate voltage.

As we discussed in response to comments 4 and 5 by Reviewer #2, analysis of our data suggests that the changes in superfluid stiffness and the normal state resistance are mainly driven by changes in the electronic disorder that affects the mobility and the electron scattering time .

However, we can only speculate about the mechanisms that cause and change the electronic disorder in our devices. Both the SiO₂ substrate and the ionic gate are potential sources of electrostatic disorder, as well as pointed out by the reviewer below, intrinsic defects in the MoS₂.

If disorder from SiO₂ substrate plays an important role, the conducting layer at the bottom of the flake could screen disorder from the SiO₂ substrate with increasing strength as the density in this layer increases with backgate voltage. Since the formation of this bottom layer at positive backgate voltages is supported by Refs. [5,6] of this letter, this is an appealing mechanism in that regime. At negative backgate voltages, possibilities include that the self-screening properties of the top layer changes sufficiently due to changes in its density, or the shape of confinement potential could get slightly modified in turn changing the disorder. However, we have not found a convincing way to model these effects, and thus the mechanism(s) that affect the disorder in the device remain unclear as we mention in the manuscript.

The reviewer suggests an interesting additional mechanism: the backgate might induce a slight motion of the ions in the frozen ionic liquid. This would lead to a large change in the electron density and potentially disorder. However, the ions near the surface of the MoS₂ flake are positively charged. When the backgate is set to positive voltage, this would tend to push them away from the conducting layer, thus reducing the electron density, which would lead to a reduced superfluid response, in contrast to the increase we observe. It is also worthwhile noting that our ionic gate freezes at approximately 220 K. Therefore, we are far below the temperature when the ions are readily mobile, and we could not find reports in the literature of ions moving at cryogenic temperatures.

We have edited our discussion of the role of the backgate and possible sources of disorder in our devices in line with this response.

In nearly all cases, the resistive superconducting transitions are nearly completed before the susceptibility sees the superconductivity. The authors do not really address this point. If this is

because there are special paths that dominate the transport, is it then reasonable to use the sheet resistance from the normal state in the superfluid density calculations?

Thank you for giving us the opportunity to discuss this aspect of the superconducting transition in our samples. Indeed, the resistive transition is nearly complete at the temperature T at which the superfluid response of the devices exceeds our noise floor. We specifically arranged Fig. 2 so the reader could easily see the relationship between the magnetic response and transport.

The behavior that we observe is consistent with what is expected for a Berezinski-Kosterlitz-Thouless transition in a two-dimensional superconductor. Both theory (Ref. [7] in this letter) and previous experiments on bulk films (Ref. [8] in this letter) have shown that the resistance is nearly completely suppressed before a significant diamagnetic response emerges. Experimentally, this was observed even in early experiments using mutual inductance measurements to study the BKT transition (Ref. [8] of this letter)). From theory, the sheet conductivity and the diamagnetic response are expected to be proportional to the square of the correlation length (see Ref. [7] of this letter). Between the mean-field T_C and T_{BKT} , the correlation length is finite and grows rapidly as T approaches T_{BKT} . Therefore, one might expect that as the resistance starts decreasing, weak fluctuating diamagnetism should appear. However, we expect this fluctuating diamagnetism to be below our noise floor given the calculation in Ref. [7] of this letter.

The reviewer brings up an alternative scenario in which an isolated superconducting path forms between the contacts of the transport measurement. In that case, the resistive transition would be dominated by the formation of a superconducting path through the device. The resistance would start to decrease when the first superconducting filament forms, and the resistance drop would end when the path is completed. Narrow superconducting regions have a weak superfluid response due to weak and confined screening currents. We would therefore only observe a superfluid response once the filament has grown into a superconducting region of finite area, which would likely occur below the temperature where the resistive transition is completed. In this scenario, the temperature at which the resistive drop ends and the onset of the superfluid response do not have to be as strongly correlated, in contrast to our data. Further, the observation of the universal jump in the superfluid response in device A would be surprising. However, we also note that we cannot detect the presence of very weak diamagnetism above T_C regardless of whether it emerges from a superconducting filament, small superconducting regions, or the expected fluctuation diamagnetism above a BKT transition.

We briefly comment on the question on whether comparing the sheet resistance from the normal state to the superfluid response is reasonable. As we argued above, we don't believe that our data is strongly suggestive of special paths forming in the sample. By using the normal state resistance of the device to compute the sheet resistance, we follow common practice of transport measurement done on vdW devices of similar size in which the mobility is computed under the same assumptions. We certainly believe that the analysis in Fig. 3g,h is reasonable to perform. The fact that it yields values of the superconducting gap consistent with previous tunneling results (see Ref. [9] of this letter) is an indication that it is meaningful.

Finally, we would like to emphasize (as we did in our response to reviewer #2), that despite using a local probe, our measurement is closer to a bulk measurement than a local measurement for vdW

devices. The SQUID is placed a few micrometers above the sample, and the field coil is of comparable size to the sample. This means that the magnetic response measurement probes a substantial volume fraction of the entire device, which is approximately 15 micrometers in diameter. We note that even with a substantially smaller probe this would be true, because the ionic gel imposes a minimum distance between the sensor and the device. We believe that the main achievement in our work is to see the superfluid response at all, not to image it with meaningful spatial resolution, and we achieve this by the enhanced sensitivity that a local probe provides in the case of magnetic response measurements. Because we can scan the SQUID, we can answer a few questions by imaging (for example the presence of pronounced line defects), but there are clear limitations in resolving subtle spatial variations. We also limited the number of images taken in these experiments, because the samples are generally fragile and the risk of damaging them during scanning was significantly enhanced.

We have edited the revised manuscript to make a few key points from this discussion clearer:

- **We highlight that our measurement is closer to a bulk measurement than a local measurement of our devices given the measurement geometry in the introduction.**
- **We note that we cannot exclude weak diamagnetism above T_{χ} , regardless of the mechanism.**
- **In the supplement, we include an image series taken on device C (discussed in the supplement), which is the only device that we have images of as a function of temperature. These show that the spatial structure of the superfluid response is largely temperature independent.**

Regarding the effect of disorder - the authors invoke extrinsic disorder like the SiO₂ substrate. In my opinion, a much simpler effect is the change in the potential landscape as a function of chemical potential. It is well known that the TMD semiconductors show poor transport properties near the band edge. The authors could comment on this possibility and perhaps include appropriate references.

We thank the reviewer for the suggestion to consider intrinsic origins of disorder in these materials. Indeed, even when MoS₂ is placed on clean hBN substrates its mobility remains modest, demonstrating that intrinsic disorder, of the type the reviewer mentions, is important. However, substrate disorder is well established as an important mechanism as well, given the hBN substrate reports we cite (Ref. [10] in this letter) and previous transport work on the electrostatics of this system. **In the revised manuscript, we have included intrinsic disorder as an additional possible source of disorder in our devices.**

Once again, we would like to thank all the reviewers for your time and valuable comments on our manuscript. We look forward to hearing your feedback, and sincerely hope that we have addressed all your questions and comments.

References:

- [1] Ye, J., Zhang, Y., Akashi, R., Bahramy, M. S., Arita, R. & Iwasa, Y. Superconducting Dome in a Gate-Tuned Band Insulator. *Science* 338, 1193–1196 (2012)
- [2] J. M. Lu, O. Zheliuk, I. Leermakers, N. F. Q. Yuan, U. Zeitler, K. T. Law & J. T. Ye. Evidence for two-dimensional Ising superconductivity in gated MoS₂. *Science* 350, 1353–1357 (2015)
- [3] Kogan, V. G. & Kirtley, J. R. Meissner response of superconductors with inhomogeneous penetration depths. *Physical Review B - Condensed Matter and Materials Physics* 83, 1–9 (2011)
- [4] Costanzo, D., Jo, S., Berger, H. & Morpurgo, A. F. Gate-induced superconductivity in atomically thin MoS₂ crystals. *Nature Nanotechnology* 11, 339–344 (2016).
- [5] Chen, Q. H., Lu, J., Liang, L., Zheliuk, O., Ali El Yumin, A. & Ye, J. Continuous Low-Bias Switching of Superconductivity in a MoS₂ Transistor. *Advanced Materials* 30, 1–6 (2018).
- [6] Chen, Q. H., Lu, J. M., Liang, L., Zheliuk, O., Ali, A., Sheng, P. & Ye, J. T. Inducing and Manipulating Heteroelectronic States in a Single MoS₂ Thin Flake. *Physical Review Letters* 119, 1–6 (2017)
- [7] Halperin, B. I. & Nelson, D. R. Resistive transition in superconducting films. *Journal of low temperature physics* 36, 599 (1979).
- [8] A. F. Hebard & A. T. Fiory. Evidence for the Kosterlitz-Thouless Transition in Thin Superconducting Aluminum Films. *Phys. Rev. Lett.* 44, 291 (1980)
- [9] Costanzo, D., Zhang, H., Reddy, B. A., Berger, H. & Morpurgo, A. F. Tunnelling spectroscopy of gate-induced superconductivity in MoS₂. *Nature Nanotechnology* 13, 483–488 (2018)
- [10] Cui, X., Lee, G. H., Kim, Y. D., Arefe, G., Huang, P. Y., Lee, C. H., Chenet, D. A., Zhang, X., Wang, L., Ye, F., Pizzocchero, F., Jessen, B. S., Watanabe, K., Taniguchi, T., Muller, D. A., Low, T., Kim, P. & Hone, J. Multi-terminal transport measurements of MoS₂ using a van der Waals heterostructure device platform. *Nature Nanotechnology* 10, 534–540 (2015)

REVIEWERS' COMMENTS

Reviewer #1 (Remarks to the Author):

The authors have adequately addressed the referee's comments. The manuscript is suitable for publication in Nature Communications.

Reviewer #2 (Remarks to the Author):

The authors have convincingly addressed most of my comments. I think the manuscript is now ready for publication in Nature Communications.

Reviewer #3 (Remarks to the Author):

Thanks for the authors' effort to well address my questions. I'm happy to recommend for publication.